# *Myrica esculenta* Buch.-Ham. ex D. Don: A Natural Source for Health Promotion and Disease Prevention

**DOI:** 10.3390/plants8060149

**Published:** 2019-05-31

**Authors:** Atul Kabra, Natália Martins, Rohit Sharma, Ruchika Kabra, Uttam Singh Baghel

**Affiliations:** 1I.K. Gujral Punjab Technical University, Kapurthala, Punjab 144603, India; atul.kbr@gmail.com; 2Kota College of Pharmacy, Kota Rajasthan 325003, India; ruchika.p88@gmail.com; 3Faculty of Medicine, University of Porto, Porto 4200-319, Portugal; 4Institute for Research and Innovation in Health (i3S), University of Porto, Porto 4200-135, Portugal; 5Central Ayurveda Research Institute for Drug Development, CCRAS, Ministry of AYUSH, Government of India, Bidhannagar, Kolkata, West Bengal 700091, India; dhanvantari86@gmail.com

**Keywords:** *Myrica esculenta*, kaphal, ethnomedicinal, phytoconstituents, conservation, micropropagation, pharmacology

## Abstract

*Myrica esculenta* (Myricaceae) is a popular medicinal plant most commonly found in the sub-tropical Himalayas. It is widely used in folk medicine to treat several ailments such as asthma, cough, chronic bronchitis, ulcers, inflammation, anemia, fever, diarrhea, and ear, nose, and throat disorders. Due to its multidimensional pharmacological and therapeutic effects, it is well recognized in the ayurvedic pharmacopeia. However, the recent upsurge in *M. esculenta* use and demand has led to illicit harvesting by the horticultural trade and habitat loss, pushing the plant to the brink of extinction. Thus, the present review aims to provide updated information on *M. esculenta* botany, ethnomedicinal uses, phytochemistry, pharmacological effects, toxicity, and conservation methods, as well as also highlight prospective for future research. Particular emphasis is also given to its antioxidant potential in health promotion. In-depth literature was probed by searching several sources via online databases, texts, websites, and thesis. About 57 compounds were isolated and identified from *M. esculenta,* and the available reports on physicochemical parameters, nutritional and high-performance thin-layer chromatography analysis of bioactive plant parts are portrayed in a comparative manner. Friendly holistic conservation approaches offered by plant biotechnology applications, such as micropropagation, germplasm preservation, synthetic seed production, and hairy root technologies are also discussed. Nonetheless, further studies are needed to propose the mechanistic role of crude extracts and other bioactives, and even to explore the structure–function relationship of active components.

## 1. Introduction

Genus *Myrica* is a large group comprising more than 97 species in the Myricaceae family. This family contains woody plants native to the subtropical and temperate zones of the earth [1]. Plant species of this genus are distributed in China, Taiwan, Japan, Western Highland of Cameroon, North America, South Africa, Australia, Brazil, Ethiopia, Nepal, and India [2,3,4,5]. Specifically, *Myrica esculenta* Buch.-Ham. ex. D. Don named ‘Hairy Bayberry’ and widely known as Kaiphal or Kataphala in the Indian subcontinent, is broadly used in Ayurveda (traditional Indian system of medicine) [6,7,8]. But this plant also has other synonyms, such as *Myrica sapida* Wall. and *Myrica farqhariana* Wall. [5,9,10]. *Myrica* plants grow well in nitrogen-depleted soils, mixed forests, agricultural and marginal lands [1,11]. *Morella esculenta* (Buch.-Ham. ex. D. Don) I.M. Turner is the newly accepted name for *Myrica esculenta* Buch.-Ham. ex.D. Don, and the later name is treated as a basionym of *Morella esculenta*. Taxonomical classification of *Myrica esculenta* is Kingdom: Plantae; Phylum: Tracheophyta; Class: Magnoliopsida; Order: Fagales; Family: Myricaceae; Genus: *Morella* [12].

*M. esculenta* is known for its edible fruits and other by-products. Indeed, its fruits have been a potential income generating source for the local tribes of the Meghalaya and sub-Himalayan region [13,14]. It is likewise known by a variety of names, such as “Katphal” in Sanskrit, “Kaiphal” in Urdu, “Nagatenga” in Assam, ‘Soh-phi’ in Khasi and ‘Box myrtle’ in English [1,15,16,17]. All the parts of the *M. esculenta* plant have huge nutritional and therapeutic importance. Fruits are used for syrups, jams, pickles, and preparation for refreshing drinks [14]. Traditionally, its bark, roots, and leaves are used for the treatment of various ailments and disorders [3,5]. Besides its traditional uses, bark is also used for making paper and ropes [18]. In addition, *M. esculenta* fruits and roots are used as an active botanical ingredient in numerous ayurvedic formulations (Table 1).

More recently, its numerous ethnomedicinal uses led researchers to explore *M. esculenta* phytochemistry further. For instance, tannins extracted from its bark are used as a dyeing agent [6]. Indeed, the presence of distinct bioactive compounds, such as alkaloids, flavonoids, glycosides, tannins, terpenoids, saponins, and volatile oils [8,21], has been increasingly reported as related to its pharmacological effects. For example, crude extracts and isolated compounds from *M. esculenta* exhibit both in vivo and in vitro pharmacological activities. Local tribes use the tree for timber, fuel, fodder, wood, likewise as used for tanning and getting yellow colored dye [22,23,24,25]. In spite of being a useful tree, the cultivation of the plant is incredibly restricted, and most of the traditional and commercial uses of *M. esculenta* rely solely on collections from the wild sources by endemic people [26]. Thus, wild sources of the species are underneath impending danger of extinction due to the increase in urbanization, overharvesting, negligence of sustainable use, and over-exploitation of forests and wastelands for industrial uses [27]. Due to the over-exploitation of the natural habitat, limited geographical prevalence and the unresolved problems inherent in seed vitality and germination, alternative propagation and conservation approaches are desperately needed to avoid the possible extinction of this vital species [8]. This species is fundamentally the same as *M. rubra*, which is ordinarily found in China and Japan. However, *M. esculenta* contains fruits smaller than about 4–5 mm compared to the *M. rubra* fruits (12–15 mm) [28]. Although information on phenolic content and antioxidant activity of the fruit extract, juice, jam and marc of *M. rubra* [19,20,29,30,31,32] is available, this information is lacking for *M. esculenta*. Previous reviews have suggested that myricetin is obtained mainly by members of the Myricaceae family [33,34] and is a key ingredient in many foods, besides to be used as a food additive due to its antioxidant activity and ability to protect lipids from oxidative damage [35]. It is one of the key ingredients of various foods and beverages. The compound has a wide range of potentialities that include strong antioxidant, anticancer, antidiabetic and anti-inflammatory effects, and can protect a wide variety of cells from in-vitro and in vivo lesions [36]. It was first isolated in the late eighteenth century from the bark of *Myrica nagi* Thunb. (Myricaceae), harvested in India, as light-yellow crystals [37].

In this sense, this review investigates the relevant information on botanical description, ethnomedicinal uses, phytochemistry, antioxidant activity, pharmacological activity and toxicity, along with conservation of *M. esculenta.* Its critical aspects as a natural source of antioxidant compounds for health promotion and disease prevention are also raised.

## 2. Research Methodology

The research methodology adopted for the selection of articles for this review is stipulated as flow chart in Figure 1.

## 3. Botanical Description

### 3.1. Habitat

*M. esculenta* is a small, evergreen, dioecious tree [7]. It is native to Republic of India and usually available in the mountain ranges from Ravi eastward to Assam, as well as Arunachal Pradesh, Meghalaya, Sikkim, Assam, Nagaland, Manipur, Mizoram in Khasi, Jaintia, Kamarupan and the Lushai hills (Figure 2) at an elevation of 900–2100 m [26,38,39,40,41,42,43]. This species is additionally found in Nepal [44,45], China [6], Vietnam [46], Sri Lanka [47], Sylhet (Bangladesh), Pakistan and Japan, Asian country islands, Himalayas [48,49,50] and the hills of Burma [3,5].

### 3.2. Morphologicaland Microscopical Characteristics

Morphological characterization of *M. esculenta* plant and its parts (Figure 3a–d) describes that it is a small moderate sized evergreen woody tree with a height of 3–15 m. Its leaves are lanceolate, obovate, with diameter 9 × 3 cm, and lower surface shows light green; upper surface dark green in appearance [39,41].

Transverse sections of the leaf showed that the upper and lower epidermis consist of single-layered polygonal cells that cover the mucilaginous cuticle; vein islet and vein termination were 9–11 and 13–15, respectively [21,51]. Transverse sections of matured stem bark revealed multi-layered cork, made of rectangular, tangentially elongated, thin-walled cells, whereas the secondary cortex contained rectangular-polygonal parenchymatous cells with oval shaped starch grains [38,39,52,53,54].

## 4. Ethnomedicinal Uses

*M. esculenta*, a conventional ayurvedic plant, is used by different native population groups in multiple ways because of the various therapeutic uses of its bark, roots, fruits, leaves and flowers (Table 2) [20,49,55,56].

Apart from these ethnomedicinal uses, various fruit industries in Himalaya used its fruits for making syrup, jam, and squash [70]. The Khasi tribe of Meghalaya uses its bark as fish poison while the extracted tannin from its bark is use as a tanning and dyeing agent [71]. Local peoples in Arunachal Pradesh use this tree for timber and fuel [22].

## 5. Physiochemical and Nutritional Analysis

Numerous physiochemical and nutritional parameters of *M. esculenta* have been studied, as shown in Table 3 and Table 4 [22,72,73,74]. 

## 6. Phytochemistry

Phytochemical screening performed on leaves, stem bark, bark, fruits and fine branches of *M. esculenta* revealed several active phytoconstituents such as tannins, phenolic acids, flavonoids, terpenes, glycosides, steroids, volatile oils, and amino acids [8,21]. These phytoconstituents have shown a wide variety of pharmacological effects. HPTLC profiles of various extracts from different *M. esculenta* plant parts are presented in Table 5. The mobile phase used to develop the HPTLC chromatogram for n-hexane, ethyl acetate and ethanol extracts of stem bark and fine branches were toluene: ethyl acetate (5:5 *v*/*v*), toluene: ethyl acetate (7:3 *v*/*v*) and toluene: ethyl acetate: formic acid (5:5:0.5 *v*/*v*) [8] respectively, while for leaves, ethyl acetate, methanol and aqueous extracts of leaves toluene: ethyl acetate (7:3) was used [21].

### 6.1. Tannins and Phenolic Acids

*M. esculenta* bark present gallic acid; epigallocatechin 3-*O*-gallate; epigallocatechin-(4β→8)-epigallocatechin3-*O*-gallate;3-*O*-galloyl-epigallocatechin-(4β→8)-epigalloc-atechin3-*O*-gallate along with the hydrolyzable tannin castalagin [6,75]. Reversed-phase high-performance liquid chromatography analysis of fruit extract showed the presence of catechin;gallic acid; chlorogenic acid and *ρ*-coumaric acids [76]. Ethyl-β-D-glucopyranoside; 3-hydroxybenzaldehyde; isovanillin; 4-(hydroxymethyl)-phenol; 4-methoxybenzoic acid have been identified in leaves [77]. LC-MS analysis of fruit extract also indicated the presence of bioactive compounds, such as gallic acid and ferulic acids [78].

### 6.2. Flavonoids

Myricetin was also reported in leaves, fruits, and stem bark [8,46,56], whereas quercetin was found only in leaves [79]. 

Two flavonoid glycosides flavone 4′-hydroxy-3′,5,5′-trimethoxy-7-*O*-β-D-glucopyranosyl(1→4) -α-L-rhamnopyranoside and flavone 3′,4′-dihydroxy-6-methoxy-7-*O*-α-L-rhamnopyranoside were found in the leaves [79], while myricetin-3-*O*-(2″-*O*galloyl)-α-L-rhamnopyranoside and myricetin 3-*O*-(2″-*O*-galloyl)-α-L-rhamnopyranoside were revealed in bark [78]. Myricetin 3-*O*-rhamnoside (myricitrin) was accounted in both *M. esculenta* bark, and leaves [46,77,79,80]. 

### 6.3. Terpenes

#### Monoterpenoid

Myresculoside (4-hydroxy-1,8-cineole 4-O-β-dapiofuranosyl (1→6)-β-D-glucopyranoside) were reported in the leaves of *M. esculenta* [46].

### 6.4. Triterpenoids

Numerous triterpenoids such as lupeol; Oleanolic acid;trihydroxytaraxaranoic acid; dihydroxytaraxerane; dihydroxytaraxaranoic acid; tetrahydroxytaraxenoic aci; 3-epi-ursonic acid; arjunolic acid were reported in bark and leaves of *M. esculenta* [46,75,81,82].

### 6.5. Volatile Compounds

The volatile compounds identified in leaves [83] were nerolidol; α-pinene; α-selinene; β-caryophyllene; β-selinen; α-caryophyllene; α-cadinol; linalool; whereas in bark were n-hexadecanol; eudesmol acetate and n-octadecanol [82].

### 6.6. Proanthocyanidins

*M. esculenta* bark revealed the presence of proanthocyanidins, such as proanthocyanidin acetate; proanthocyanidin methyl-ether and prodelhinidin [84,85].

### 6.7. Diarylheptanoids

*M. esculenta* bark, leaves and root exhibited the presence of diaylheptanoids. Myricanol and myricnone were reported in bark [6,84,86] and leaves, whereas 13-oxomyricanolwas reported in root [86], 5-*O*-β-D-glucopyranosylmyricanol was accounted in leaves [45], and 16-bromomyricanol was identified in bark [86].

### 6.8. Steroids

β-rosasterol; daucosterol; β-sitosterol-β-D-glucopyranoside were identified in leaves [77,80] where as taraxerol, stigmasterol were found in bark [74,80,87]. β-sitosterol was identified in both *M. esculenta* leaves [77,80] and bark [81,88]. Other miscellaneous compounds, such as amino acids; 1-ethyl-4-methylcyclohexane, myo-inositol, methyl-d-lyxofuranoside, 2-furancarboxyaldehyde, 2,5-furandionedihydro-3-methylene, furfural, oxirane were also reported in *M. esculenta* fruits [73,78].

The structures of some important bioactive phytoconstituents reported in *M. esculenta* plant are presented in Figure 4.

## 7. Pharmacological Profile

Extracts from *M. esculenta* possess a broad spectrum of pharmacological activities. Previous research revealed that phenolic compounds are highly active antioxidants, and such antioxidant-rich botanicals offer promising potential in the management of degenerative ailments. Phenolic compounds are secondary metabolites synthesized in plants in response to environmental stresses such as attacks from pathogens and insects, UV radiation, and injuries [5,6,7]. These phytochemicals have the ability to eliminate hydroxyl radicals [89], superoxide anion radicals [90], lipid peroxyl radicals [91] and even to chelate metals, besides to play a vital role in the stability of food products, as well as in the defense mechanisms of biological systems [4,8]. These molecules also prevent oxidative losses and have cytoprotective, anti-inflammatory, and adaptogenic properties. It was found that relatively high amounts of phenolic compounds are present in *M. esculenta* fruits than *M. rubra* [76]. The antioxidant activity of *M. esculenta* fruits and bark has been reported by using different antioxidant assays.

Previous research confirmed that presence of phenolic acids and flavonoids is responsible for its antioxidant potential [78,92,93,94,95,96,97,98]. But other pharmacological activities have been also reported, including analgesic [50,92,93], antiasthmatic [98,99,100,101,102], anticancer [78,103], antidepressant [61,104], antidiabetic [105], antidiarrheal [106], anthelmintic [106,107], antihypertensive [45], anti-inflammatory [50,94,108], antimicrobial [73,78,109,110,111], antipyretic [93], antiulcer [112], anxiolytic [61], chemopreventive [113], hepatoprotective [114], wound healing [59], and non-toxicity [105] effects. Simultaneously, several in vitro and in vivo studies on pharmacological profile of *M. esculenta* are under way. Scientific exploration has revealed that different types of *M. esculenta* extracts possess multiple bioactive attributes (Table 6).

Previous studies reported that the toxic impacts of methanolic extract of *M. esculenta* leaves and found no indication of lethality up to the dose of 300 mg/kg by oral route for two weeks. In any case, 2000 mg/kg of lethal impact measurements of the methanol extract were seen in Wistar rats [100]. Furthermore, intense poisonous quality examinations performed with ethyl acetate and aqueous extracts of *M. nagi* bark at three different intravenous dosages (100, 200 and 1000 mg/kg) demonstrated that the LD_50_ of the ethyl acetate and aqueous extracts in mice was 1000 mg/kg [98].

## 8. Conservation

Demolition of plant assets is an ordinary event. The current rate of eradication caused by mankind is about hundreds of time faster compared to the natural rate of elimination [117]. Due to training exercises in the Himalayan district, the existence of numerous therapeutically effective botanicals, including *M. esculenta,* are threatened with extinction. *M. esculenta* is exchanged and used most often as a conventional medication. Because of its numerous uses, application is on the rise through national and worldwide exchange, leading to the expansion of wild populaces. This has brought exceptional declines in population [118,119]. Due to its extreme overuse from regular natural surroundings, limited geographic predominance, and uncertain inalienable issues of seed practicality and seed germination, elective methodologies for spread and protection are urgently expected to evade the potential termination of this indispensable species [8,27]. The village forest council framework is a town-level establishment, and it has impressive potential for involving local communities in forest management for conservation [119]. Biotechnology offers new methods for enhancing biodiversity and biotechnological methodologies. For example, micropropagation systems have gotten more consideration and may assume a fundamental part in the foundation of hereditarily unvarying botanicals for the business. Hopefully, the advancement of coherent micropropagation conventions could ensure satisfactory availability of the *M. esculenta* plant (without forced ecological imperatives) with a consequent lessening in uncontrolled collecting weight on wild populaces [27]. Likewise, there are several highly valued traditional Indian ethnomedicinal plants having rich therapeutic potential and need immense scientific exploration and conservation strategies [120,121,122].

## 9. Conclusions and Future Perspectives

*M. esculenta* has been used for its restorative and dietary potentials, from the old-fashioned Ayurveda and Unani arrangement of solution. It is clear in this review that *M. esculenta* contains various phytochemicals, which are responsible for the therapeutic estimate of this plant. *M. esculenta*, and have been responsible for several pharmacological impacts in the treatment of different diseases, including asthma, diabetes, tumors, ulcer, tension; however, being a rich wellspring of vitamin C and polyphenolic compounds, there is a need to investigate the capacity of this plant for immunomodulatory, cardioprotective, nephroprotective, and neuroprotective movement. Although there are many analyses of chemical constituents, and the pharmacological activity has been reported for this plant, the mechanism of pharmacological action and the metabolites responsible for these activities should be studied in more detail. The population of this restorative and practical plant species is on the reverse because of excessive exploitation of woodlands and wastelands, neglect of practicable assets, poor development, and poor recovery of species in characteristic natural surroundings. Subsequently, a great opportunity has already passed to make the vital movement to expand its populace measure, efficiency, protection, and even use. 

## Figures and Tables

**Figure 1 plants-08-00149-f001:**
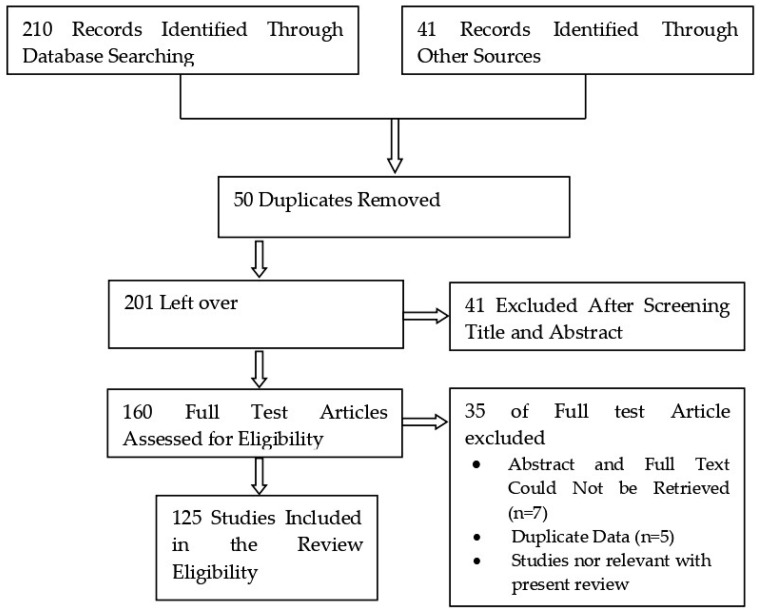
Flow diagram of research methodology.

**Figure 2 plants-08-00149-f002:**
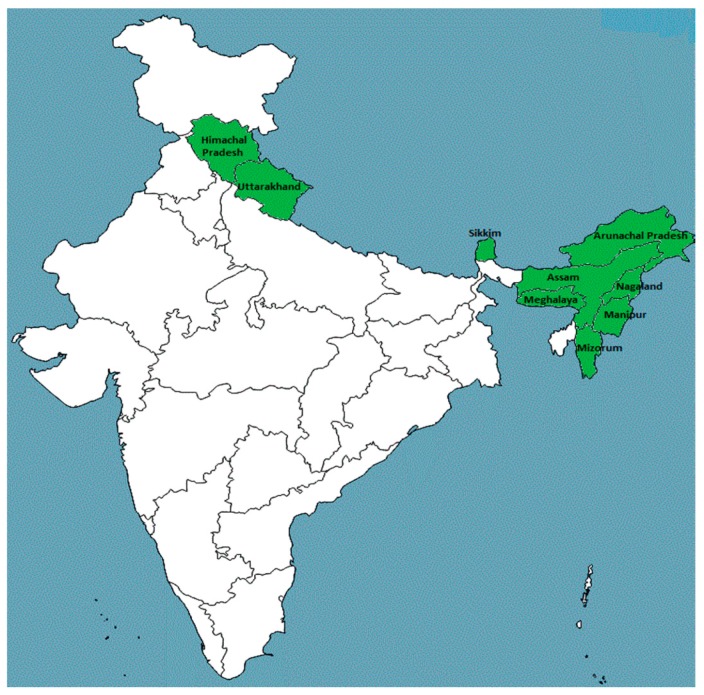
Natural distribution of *Myrica esculenta*. The shaded area represents the natural habitat of *M. esculenta* in the India.

**Figure 3 plants-08-00149-f003:**
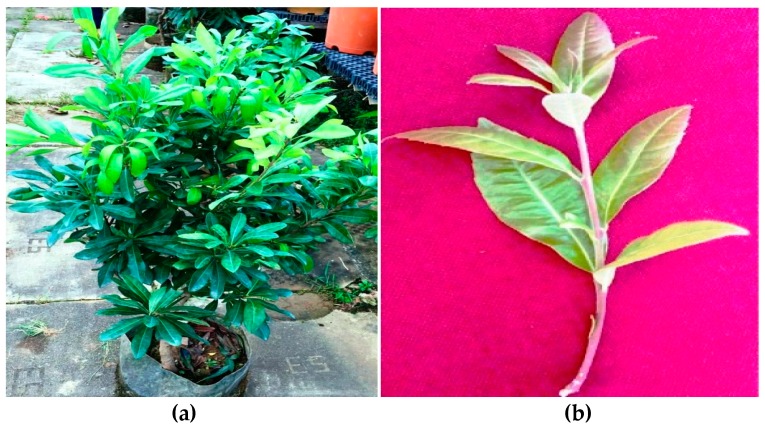
*Myrica esculenta* (**a**) Whole plant; (**b**) Leaf; (**c**) Bark; (**d**) Fruit.

**Figure 4 plants-08-00149-f004:**
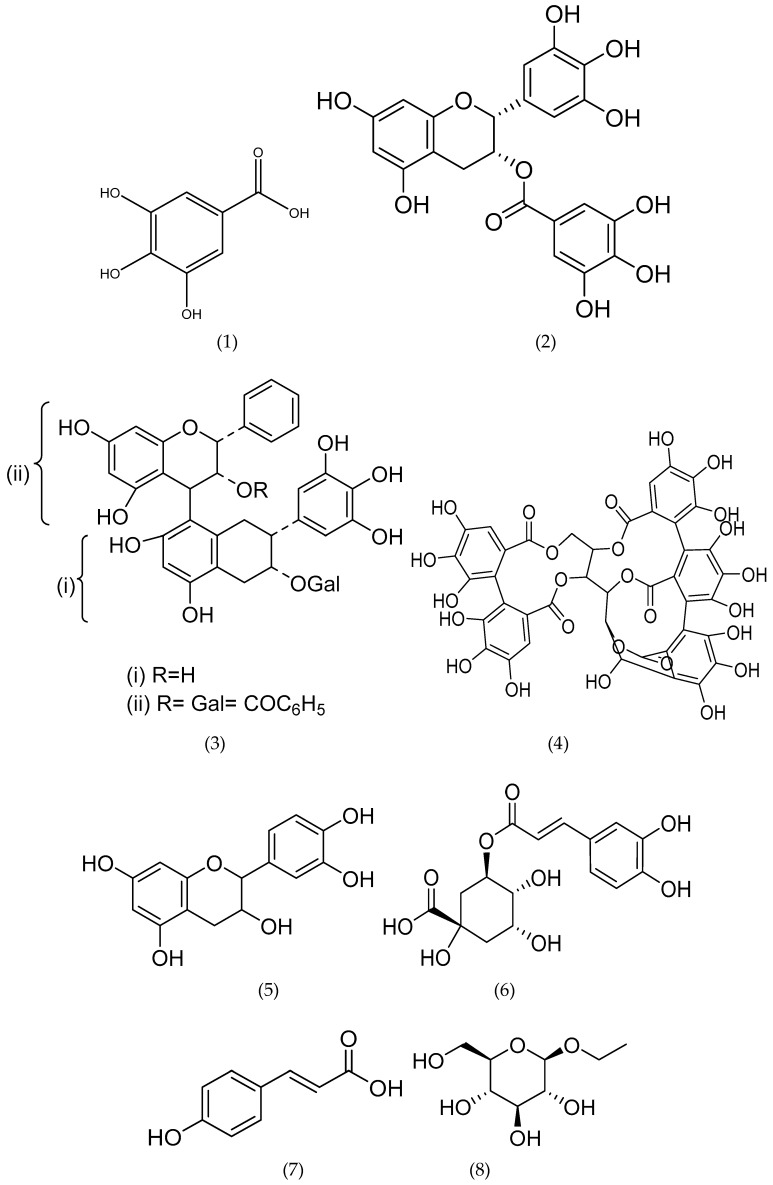
Structure of some isolated bioactive compounds from different parts of *M. esculenta.* (**1**) Gallic acid, (**2**) Epigallocatechin 3-O-gallate, (**3**) i) Epigallocatechin-(4β→8)-epigallocatechin-3-O-gallate, ii) 3-O-galloyl-epigallocatechin-(4β→8)-epigallocatechin-3-O-gallate, (**4**) Castalagin, (**5**) Catechin, (**6**) Chlorogenic acid, (**7**) *p*-coumaric acid, (**8**) Ethyl-β-D-glucopyranoside, (**9**) 3-hydroxybenzaldehyde, (**10**) Isovanillin, (**11**) Ferulic acid, (**12**) Myricetin, (**13**) i) Flavone 4′-hydroxy-3′,5,5′-trimethoxy-7-*O*-β-D-glucopyranosyl(1→4)-α-L-rhamnopyranoside, ii) flavone 3′,4′-dihydroxy-6-methoxy-7-*O*-α-L-rhamnopyranoside, (**14**) Myricitrin, (**15**) Lupeol, (**16**) Oleanolic acid, (**17**) Trihydroxytaraxaranoic acid, (**18**) Dihydroxytaraxerane, (**19**) Dihydroxytaraxaranoic acid, (**20**) Tetrahydroxytaraxenoic acid, (**21**) 3-epi-ursonic acid, (**22**) Prodelphinidin dimer, (**23**) Myricanol, (**24**) Myricanone.

**Table 1 plants-08-00149-t001:** Ayurvedic formulations of the plant with their uses and manufacturers.

Formulation	Uses	Manufacturers	References
“Chwayanprash”	Improved digestion and strength and enhanced energy	Dabur, Patanjali, Nature & Nurture Healthcare	[19,20]
“Katphaladi Churna”	Treatment of fever, throat infection, respiratory disorders, and abdominal pain	VHCA Ayurveda	[19,20]
“Pushyanuga Churna”	Treatment for bleeding disorders and candidiasis	AVN Ayurveda,Baidyanat-h	[19,20]
“Katphala Taila”	Treatment of joint pain	VHCA Ayurveda	[19,20]
“Arimedadi Taila”	Helps to relieve tooth decay and breath problem	IMIS Pharmaceuticals	[19,20]
“Mahavisagarbha Taila”	Used for vata imbalance, neuromuscular conditions	VHCA Ayurveda	[19,20]
“Bala Taila”	Treatment of vata disorders, respiratory infections and weakness	Patanjali	[19,20]
“Khadiradi Gutika”	Treatment of dental, oral, throat and tonsillar infections	Zandu	[19,20]
“Maha Vatagajankusa Rasa”	Rheumatoid arthritis, Migraine, Paralysis, Cough, Cold, Asthma	Dabur, Baidyanath, Shree Dhootapapeshwar	[19,20]
“Brihat Phala Ghrta”	Treatment of infertility	SN Pandit Ayurvedic	[19,20]

**Table 2 plants-08-00149-t002:** Ethomedicinal uses of *M. esculenta.*

Plant Part Used	Uses	Region/Tribe	References
**Leaf, fruit, root, bark**	Jaundice	Meghalaya, India	[23]
**Leaf**	Inflammation of vocal cord	Meghalaya, India	[24]
**Bark**	Antiseptic	Meghalaya, India Khasi tribe	[24]
**Fruit, bark, leaf**	Fever	Meghalaya, India Vietnam, South China	[24]
**Bark**	Anemia	Meghalaya, India Khasi tribe	[24]
**Fruit**	Refreshing drink “Um Soh-Phi”	Meghalaya, India Khasi tribe	[24]
**Bark**	Sore	Nagaland, India Zeliang tribe	[43]
**Bark**	Toothache	Meghalaya, India Khasi tribe Almora, Uttarakhand, India	[24,41,57]
**Bark**	Sprain	Far-flung village, Jajarkot, Nepal	[58]
**Flower, bark, leaf**	Inflammation, paralysis	Meghalaya, India Khasi tribe Vietnam, South China	[24,59]
**Unripe fruit**	Anthelmintic	Himachal Pradesh, India	[59]
**Fruit**	Bronchitis, dysentery	Nepalese community, Nepal	[60]
**Bark**	Mental illness	Orissa, India	[61]
**Bark**	Skin disorder	Vietnam, South China	[62]
**Bark**	Cholera	Mizoram, India	[63,64]
**Bark**	Cardiac debility, cardiac edema	Meghalaya, India	[64]
**Bark**	Carminative	Meghalaya, India Khasi tribe Mizoram, India	[22,64]
**Bark, leaf**	Asthma, chronic bronchitis, lung infection	Meghalaya, India Khasi tribe Vietnam, South China Chaubas and Syabru, Nepal	[23,63,65]
**Flower**	Earache	Meghalaya, India Khasi tribe Almor, Uttarakhand, India Himachal Pradesh, India	[24,41,49,66]
**Bark, flower, leaf, fruit**	Diarrhea, dysentery, stomach problem	Meghalaya, India Khasi tribe Almora, Uttarakhand, India Chungtia village, Nagaland, India	[24,66,67]
**Leaf**	Redness of mucosa	Chungtia village, Nagaland, India	[67]
**Fruit**	Body ache	Ukhimath block, Uttarakhand, India	[68]
**Bark, fruit**	Headache	Mizoram, India Ukhimath block, Uttarakhand. India	[64,68]
**Fruit**	Ulcer	Himalaya, India	[69]

**Table 3 plants-08-00149-t003:** Physiochemical parameters of different parts of *M. esculenta.*

Parameters	Results				References
	Leaves	Bark	Stem Bark	Small Branches	
**Extractive value (%w/w)**					[8,21,55]
Methanolic extract	28.32	38.52	23.57	5.03	
Ethyl acetate extract	25.46	21.20	NR	NR	
Aqueous extract	21.28	15.7	18.36	3.52	
**Ash Values (%w/w)**					[8,21,55]
Total ash	2.83	3.3312	1.010	1.856	
Acid insoluble ash	0.52	1.2300	0.187	0.320	
Foreign matter (% w/w)	<1%	NR	Nil	Nil	
Loss on drying (%w/w)		5	6.47	6.81	
Total phenolics mg of GAE/g d.w.	NR	NR	276.78 ± 5.36	31.24 ± 2.57	[8]
Total flavonoids mg of QE/g d.w.	NR	NR	121.68 ± 6.81	12.94 ± 1.12	[8]

**Table 4 plants-08-00149-t004:** Mineral analyses of *M. esculenta* fruits and stem bark.

Minerals (mg/g)	Fruit	Stem Bark	Reference
Calcium	4.63 ± 0.06	3.155 ± 0.18	[72,73,74]
Potassium	7.75 ± 0.11	2.939 ± 0.23	[72,73,74]
Magnesium	8.4 ± 0.20	1.061 ± 0.4	[72,74]
Sodium	0.81 ± 0.013	0.060 ± 0.03	[72,74]
Phosphorous	0.24 ± 0.25	0.030 ± 0.01	[73,74]
Manganese	0.032 ± 0.0001	NR	[72]
Iron	0.404 ± 0.0021	0.123 ± 0.16	[72,73]
Zinc	0.216 ± 0.0016	0.006 ± 0.001	[72,73]
Copper	0.004 ± 0.0002	NR	[72]
Sulphur	NR	0.277 ± 0.34	[73]

**Table 5 plants-08-00149-t005:** HPTLC profile of various extracts of different parts of *M. esculenta.*

Extract	Wavelength (nm)	Rf Value			References
		Stem Bark	Small Branches	Leaves	
n-hexane	254	0.49, 0.69, 0.88	0.49, 0.78	NR	[8]
	366	0.42, 0.51, 0.59, 0.74, 0.83,0.91	0.42, 0.51, 0.74,0.83,0.91		
Ethyl acetate	254	0.07, 0.12, 0.36, 0.47, 0.61, 0.67, 0.84	0.47, 0.67	0.15, 0.6, 0.8	[8,21]
	366	0.11, 0.15, 0.18, 0.33, 0.38, 0.55, 0.49, 0.65, 0.75, 0.85, 0.90	0.18, 0.30, 0.49, 0.65, 0.75, 0.85, 0.90	0.11, 0.22, 0.38, 0.53, 0.69, 0.82, 0.93	
Ethanol	254	0.23, 0.54	0.23, 0.54	NR	[8]
	366	0.54, 0.73, 0.84	0.25, 0.45, 0.54, 0.73, 0.84		
Methanol	254	NR	NR	0.625, 0.875	[21]
	366			0.46, 0.58, 0.81, 0.86, 0.93	
Aqueous	254	NR	NR	0.1, 0.63	[21]
	366			0.093, 0.65, 0.81	

**Table 6 plants-08-00149-t006:** Biological effects of *M. esculenta.*

Part Use	Extract/Fraction	Dose Tested/Route of Administration	Animals/Cell Lines	Experimental Models	Result	Reference
**Anti-inflammatory**						
Leaves	Methanolic	200 mg/kg, p.o.	Rat	Carrageenan-induced rat paw edema	Significant anti-inflammatory activity	[93]
Stem Bark	Essential oil	10 mL per ear	Swiss albino mice	In vitro [ear]	Significant anti-inflammatory potential	[111]
Leaves	ME- EtAC	100, 200 and 400 mg/kg, p.o.	Wistar rats	Carrageenan-induced rat paw edema	Significant anti-inflammatory activity	[50]
Bark	Ethyl acetate and aqueous	100 and 200 mg/kg, p.o.	Wistar albino rats	Carrageenan and histamine induced rat paw edema	Significant anti-inflammatory potential (EAE> AE)	[102]
**Antimicrobial**						
Stem bark	Volatile oil	10 mL	BP, SA, SE, EC, PA, CA, AN and SC	---	Significant antimicrobial potential	[111]
Bark and fruit	Methanolic and chloroform	---	---	Agar Well diffusion method	Significant antimicrobial potential (Bark> Fruits)	[109]
Fruit pulp	Ethanolic	10 and 50 mg/ml	In vitro	Disc diffusion assay	Dose dependent antimicrobial potential	[73]
Fruit	Methanolic	50 μL	SA, SE, BS, PM, EC, SE	Agar Well diffusion method	Significant potential against Pathogens	
**Antifungal**						
Fruit	Methanolic, ethanolic and aqueous	10 and 50mg/ml	*Candida albicans*, *Aspergillus flavus* and *Aspergillus parasiticus*	Disc diffusion assay	Significant antifungal activity	[73]
**Anthelmintic**						
Bark	50% Aqueous Ethanolic	12.5, 25 and 50 mg/ml	Earthworms (*Pheretima posthuma*)	---	Paralysis and death at 12.5 mg.ml	[107]
**Anticancer**						
Fruit	Methanol, acid methanol acetone and acidic acetone	66.7, 166.5, 333, 500, 667 µg f.w./100 µL culture medium	C_33_A, SiHa and HeLa cell lines	---	Acetone and acidic acetone extracts showed anticancer potential	[108]
Fruit	Methanolic	5mg/ml	HepG2, Hela and MDA-MB-231 cells	MTT assay	Moderate anticancer activity	[78]
**Chemopreventive**						
Bark	Ethanolic	2.0 mg and 4.0 mg/kg	Swiss albino mice	Cumene hydroperoxide-mediated cutaneous oxidative stress and toxicity	↑ antioxidant enzymes activity	[113]
**Antioxidant**						
Fruit	Methanolic	---	In vitro	DPPH, ABTS and FRAP assay	Significant antioxidant activity	[78]
Fruit pulp	Methanolic	0.10 ml	In vitro	DPPH, ABTS and FRAP assay	Good scavenging potential	[76]
Fruit	Aqueous methanol and acetone	100 µl	In vitro	DPPH assay	Acetone extract showed higher scavenging potential	[94]
Fruit	Methanol, acidic methanol, acetone, and acidic-acetone	---	In vitro	DPPH, ABTS, FRAP and Superoxide anion radicals scavenging assay	MeAA showed higher antimicrobial potential and MeAM and MeA intermediate	[103]
Fruit	Fruit Juice	0.2–2.0 mg/mL	In vitro	DPPH, H_2_O_2_ and NO scavenging activity	Moderate antioxidant activity	[95]
**Antidiabetic**						
Leaves	Methanolic	50,100 and 150 mg/kg, p.o.	Albino wistar rats	STZ induced diabetes	Significant anti-dyslipidemic effect at 150 mg/kg and maintain blood glucose level	[105]
**Hepatoprotective**						
Polyherbal formulation (Herbitars)	---	50 and 100 mg/kg	Wistar rat	CCl_4_ induced hepatotoxicity	Extract ↓TBARS, ↑SOD, CAT, GSH	[114]
**Antidepressant**						
Bark	Methanolic	300, 500 mg/kg, p.o.	Albino mice	Open field test, cage-crossing test, head-dip test, rearing test, traction test, forced swimming test	Significant antidepressant activity	[104]
**Anxiolytic**						
Bark	Ethanolic	100, 200 and 400 mg/kg	Rats	Tail suspension test	Significant and dose dependent anxiolytic activity	[61]
				Forced swimming test		
**Antihypertensive**						
Leaves	Methanolic	100 mM	In vitro	ACE inhibitory activity	Potent ACE inhibition potential	[45]
**Antiasthmatic**						
Bark	Ethanolic	75 mg/kg, p.o.	Guinea pig	Acetylcholine induced bronchospasm	Protection against bronchospasm and anaphylaxis	[98]
Bark	Ethanolic	75 mg/kg, p.o.	In vitro	Guinea pig tracheal strip	↓pD2 value of histamine and acetylcholine	[98]
Stem bark	Ethanolic	150 mg/kg, p.o.	Guinea pig	Histamine induced bronchospasm	↓TLC and DLC	[115]
Stem Bark	Ethanolic	75 and 150 mg/kg, p.o.	Mice	Acetic acid induced vascular permeability and allergic pleurisy		[99]
Stem bark	Aqueous extract	27 & 54mg/kg p.o.	Guinea pig	histamine induced bronchospasm	Significant antiasthamtic potential	[100]
			In vitro	Guinea pig tracheal chain		
Bark	Polar, non-polar and methanolic	200 mg/kg, p.o.	Rat and in vitro	Acetylcholine induced bronchospasm in conscious guinea pigs; acetylcholine induced contraction on isolated guinea pig tracheal chain preparation; compound 48/80 induced mast cell degranulation using rat; and trypsin and egg albumin induced bronchospasm	PE showed higher potential than NPE and ME	[101]
**Antiulcer**						
Bark	Ethanolic	100 and 200 mg/kg	Albino rat	Pyloric ligation induced ulcer	↓level of GV, FA, LPO and GSH and ↑ CAT, nitrate and MPO↓level of GV, FA, LPO and GSH and ↑ CAT, nitrate and MPO	[112]
**Antidiarrheal**						
Leaves	Ethanolic	250 and 500 mg/kg, p.o.	Mice	Castor-oil induced diarrhea	Significant antidiarrheal activity	[106]
**Antipruritic**						
Stem bark	Aqueous	150 mg/kg	Male mice	Compound 48/80-induced	Significantly decrease in scratching effect	[116]
**Analgesic**						
Fruit	Methanolic	50, 100 mg/kg, p.o.	Mice	Eddy’s hot plate method	Significant analgesic activity	[92]
Leaves	ME-EtAC	100, 200 mg/kg, p.o.	Mice	Acetic acid induced writhing and tail immersion assay	Significant response at 200 mg/kg	[50]
Leaves	Methanolic	200 mg/kg, p.o.	Mice	Acetic acid induced writhing	54.56% inhibition of writhing	[93]
**Antipyretic**						
Fruit	Methanolic	50 and 100 mg/kg	Mice	Yeast induced pyrexia in mice	Significant antipyretic effect at 100 mg/kg	[92]
**Wound healing**						
Bark	Aqueous	Ointment (100 mg/500mm^2^)	Albino rats	Wound excision and incision	Significant wound healing potential	[59]

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
