# Peer review of "Myrica esculenta Buch.-Ham. ex D. Don: A Natural Source for Health Promotion and Disease Prevention"

_plants, 2019, doi:10.3390/plants8060149_

Round 1

Reviewer 1 Report

Suggestions:

Line 2. Myrica esculenta Buch.-Ham.  ex D. Don:

Space missing between esculenta and Buch., ex (Not Ex.) and D. Don (Not D. Don.)

Line 23.  esculenta (Not Esculenta)

Introduction: It is suggested to insert one sentence as follows:

Morella esculenta (Buch.-Ham. ex D. Don) I.M. Turner is the new accepted name for Myrica esculenta Buch.-Ham. ex D. Don, and the latter name is treated as basionym of Morella esculenta (www.catalogue oflife.org; www.gbif.org).

Line 44 and 55: Delete two synonyms: Myrica nagi Hook. f. and Myrica integrifolia Roxb. and retain two synonyms: Myrica sapida Wall. and Myrica farquhariana Wall. (See: q is missing between q and h, and it should be Wall. (not Wall).

Line 128. In India (Not in the India)

Line 158. Insert space before 'are'

Line 159. Insert space before A

Line 160. Insert space before Vietnamese

Line 287. In vitro (not invitro), insert space between in and vitro

Author Response

Authors are thankful for your valuable suggestion on manuscript. 

Reviewer 2 Report

Introduction. There are general mistakes about the distribution of the genus and Family. The botanical identification of the plant is not clear. There is confusion about the names.The synonyms mentioned are not contrasted in a General Database as Tropicos (www.tropicos.org). 

Research Methodology. The description of the research method is not detailed, nor reproducible. A published protocol for review papers (vg. Prisma 2009 Flow Diagram) is not followed.

Botanical description. Redundant information with too many paragraphs, too big distribution map vs. very small plant images (=figures 1 & 2 can be improved). The anatomic description does not offer significant information for this review

Ethnomedicinal Uses, should include a Table with the information summarized upon Categories of Uses, upon any published Ethnobotanical classification system

Physiochemical and Nutritional analysis. It should be summarized because it does not offer so valuable information for this paper.

Phytochemistry. Chemical Formula of the components must be included, in a Table at Appendix. Blocs 6.2 and 6.3 must be considered together. Blocks 6.4, 6.5, 6.7  must be reviewed  and Terpenes must be organized by groups.

Pharmacological profile. Table 5 should be reorganized. Biological effects should be ordered not alphabetically but upon Physiological criteria (Apparatus) or a general clinical classification

Safety evaluation. Lines 290 to 293 should be better deleted, because they can induce misunderstanding.

Author Response

Authors are very thankful for your valuable suggestion and improvement of our manuscript. 

Reviewer 3 Report

In this review the authors describes key aspects of Myrica esculenta, a plant tipical of sub-tropical Himalayas. with many pharmaceutical properties. The review is well written and of general interest. As major considerations before acceptance:

1) improve quality of figure 1 and 2

2) Paragraph 6; Please add some figures with chemical structure of representative compounds.

3) Line 236 myricetin, and myricitrin were indicated as terpenes

Author Response

Authors are very thankful for your suggestion. 

Round 2

Reviewer 2 Report

Figure 2 is too big. It should be reduced

Figure 3 a& b shoud be improved

Table 2 should be changed to the Table Format of the Journal (see Table 3)

Numeration of biblography of Table 2 seems not be actualized

Figure 4 should be improved

Table 6 shoud be summarized as much as possible (at least in te Results column)

-Line 287 M esculenta is not well spelled and format will be cursive

Author Response

Thanks for your valuable suggestions on our manuscript 

Reviewer 3 Report

I suggest acceptance on the present form

Author Response

Thanks for your satisfactory remark on manuscript.
